# Main Molecular Pathways Associated with Copper Tolerance Response in *Imperata cylindrica* by de novo Transcriptome Assembly

**DOI:** 10.3390/plants10020357

**Published:** 2021-02-13

**Authors:** Catalina Vidal, Giovanni Larama, Aníbal Riveros, Claudio Meneses, Pablo Cornejo

**Affiliations:** 1Centro de Investigación en Micorrizas y Sustentabilidad Agroambiental, CIMYSA, Universidad de La Frontera, Avda. Francisco Salazar, Temuco 4780000, Chile; c.vidal04@ufromail.cl; 2Programa de Doctorado en Ciencias de Recursos Naturales, Universidad de La Frontera, Avda. Francisco Salazar, Temuco 4780000, Chile; 3Centro de Modelación y Computación Científica, Universidad de La Frontera, Avda. Francisco Salazar, Temuco 4780000, Chile; giovanni.larama@ufrontera.cl; 4Centro de Biotecnología Vegetal, Facultad de Ciencias de la Vida, Universidad Andrés Bello, República 330, Santiago 8370186, Chile; a.riverosorellana@uandresbello.edu (A.R.); claudio.meneses@unab.cl (C.M.); 5FONDAP Center for Genome Regulation, Facultad de Ciencias de la Vida, Universidad Andrés Bello, República 330, Santiago 8370186, Chile

**Keywords:** Cu tolerance, Cu toxicity, de novo transcriptome, metallophyte, RNA-Seq

## Abstract

The metallophyte *Imperata cylindrica* inhabits copper (Cu) polluted soils in large areas from Central Chile. Here, we subjected clonal vegetative plantlets to 300 mg Cu kg^−1^ of substrate for 21 days to identify the main molecular pathways involved in the response to Cu stress. Transcriptomic analyses were performed for shoots and roots, with and without Cu supply. RNA-Seq and de novo transcriptome assembly were performed to identify the gene response associated with molecular mechanisms of Cu tolerance in *I. cylindrica.* De novo transcriptome revealed a total of 200,521 transcripts (1777 bp) comprising ~91% complete ultra-conserved genes in the eukaryote and Plantae database. The differentially expressed genes (DEGs) in roots were 7386, with 3558 of them being up-regulated and the other 3828 down-regulated. The transcriptome response in shoots was significantly less, showing only 13 up-regulated and 23 down-regulated genes. Interestingly, DEGs mainly related with actin and cytoskeleton formation, and to a minor degree, some DEGs associated with metal transporters and superoxide dismutase activity in root tissues were found. These transcriptomic results suggest that cytoskeleton could be acting as a mechanism of Cu-binding in the root, resulting in a high Cu tolerance response in this metallophyte, which deserve to be analyzed ultra-structurally. Our study contributes to reinforcing the potential of *I. cylindrica* as a candidate plant species to be used as a phytoremediation agent in Cu-contaminated environments.

## 1. Introduction

Soil degradation originating from exacerbated mining activity has become a global environmental problem [1,2], which need to be solved in the short term. In Chile, Copper (Cu) mining has produced a strong deleterious impact, with large areas of soils contaminated with potentially toxic elements (PTEs), which highlights large amounts of Cu [3,4,5]. Although Cu is an essential micronutrient for plants and this limit is above 20–30 mg kg^−1^ [6], high concentrations in cells can be extremely toxic. The Cu toxicity is due to its ability to alter the enzyme activity and protein functionality, and also by inducing the deficiencies of other nutrients, but mainly by the generation of oxidative damage in the main macromolecules [7]. Moreover, at the rhizosphere Cu can disrupt soil characteristics and the microbiota properties, producing a loss of ecological functionality [8,9,10].

Despite the above, there are some plant populations capable to survive in soils with high levels of PTEs, known as metallophytes. These species can survive under high Cu soil levels; therefore, they can be implemented in phytoremediation processes [11,12]. Among metallophyte naturally growing in Cu-contaminated soils in central areas from Chile, *Imperata cylindrica* is a common inhabitant in the Puchuncaví Valley, which is a place recognized as one of the highest metal-polluted sites in the country. This species has been characterized in other latitudes as an iron hyperaccumulator [13,14]; nevertheless, its Cu tolerance capacity has not been deeply studied. *Imperata cylindrica* is a perennial grass widespread in some regions of Asia and other latitudes worldwide, being used for animal feed, roofing traditional houses, erosion control and also medicinal purposes [15,16]. *I. cylindrica* preferentially propagates through vegetative rhizomes [17]. This species has become an invasive weed to crops; however, in the last decades, the great potential of perennial grasses has also been evaluated as a raw material for use in the generation of biofuel and bioenergy, due to the fast growth and low cost of maintenance [18]. Some previous studies have shown the ability of *I. cylindrica* to exude high levels of low molecular weight organic acids [19] and accumulate this element in its rhizomes over 300 mg Cu kg^−1^ of dry weight. Moreover, a recent study through scanning electron microscopy and energy dispersive X-ray spectrometer detector confirmed the preferential Cu accumulation in root organs by *I. cylindrica* [20].

Among the principal mechanisms of Cu tolerance in plants, highlights are translocation by means of membrane transporters and specialized chaperones [21], immobilization in the cell wall [22], complexation with chelating agents (phytochelatins, metallothioneins, amino acids, and organic acids in the cytosol) [23,24], and sequestration into vacuole, principally in root organs [25]. However, to know and deeply understand the mechanisms involved in the Cu tolerance by metallophytes can be an interesting way to improve biotechnological process as phytoremediation of Cu-contaminated environments.

In the last decade, the use of Next Generation Sequencing (NGS) technologies has become widespread, and has contributed to the characterization of the response to various types of stress in all kinds of organisms [26,27]. This revolutionary approach allows obtaining great volume of information from the nucleotide sequence of an individual, being able to shows its gene expression response. The survival of an organism under different environmental stressors is mainly due to its ability to adapt, and this crucial role is mostly fulfilled by the process of gene regulation [28,29]. Based on the above, in this study, we aimed to characterize the Cu tolerance response of *I. cylindrica* by means of the shifts in its transcriptome and glimpse the molecular mechanisms of Cu-root accumulation based on the gene response. This knowledge can provide relevant information when considering the use of this species in Cu-phytoremediation needed for extent surfaces in north and central Chile, where Cu mining is the most important economic activity.

## 2. Results

### 2.1. Biomass Production and Localization of Cu in Tissue

The Cu applied in the growth substrate of *I. cylindrica* plants generated a significant decrease in the fresh biomass production in both shoots and roots compared to the control conditions (Figure 1A,B) after 21 days of the treatment. Moreover, the Cu accumulated was two to four times higher in the treatment with Cu addition, compared with the control without Cu addition (Figure 1C,D). The Cu localization in tissue of *I. cylindrica* through scanning electron microscopy (SEM) coupled to Energy-dispersive X-ray analysis (EDX) spectra allowed us to identify the presence of Cu in root tissues of plants exposed to treatment with this metal (Figure 2). On the contrary, the presence of Cu in shoots was not visualized irrespective of the Cu treatment.

### 2.2. Illumina Sequencing and De novo Transcriptome Assembly

To characterize the transcriptome responses of *I. cylindrica* under Cu stress, 15 cDNA libraries were sequenced, including eight constructed from shoots and seven from roots. A total of 371,336 paired-end reads were obtained. The reads were trimmed and filtered to remove low-quality reads, resulting in a total of 347,207 high-quality reads (HQR). The HQR were concatenated to assembly the *I. cylindrica* transcriptome using the Trinity software. The resulting contigs were clustered to remove those highly redundant (>95% similarity), being identified 200,521 contigs with an N50 value of 1918 bp (Table 1). The transcripts obtained were aligned to the SwissProt database, being obtained 117,238 annotated genes, corresponding to 58% of the total *I. cylindrica* contigs. To estimate the quality of the *I. cylindrica* transcriptome, we searched a set of highly-conserved orthologous genes on the Benchmarking Universal Single-Copy Orthologs (BUSCO) database. A yield of 1375 BUSCO genes, 1259 complete (91.5%), 85 fragmented (6.2%), and 31 missed (2.3%) genes were found in the *I. cylindrica* transcriptome (Table 2).

### 2.3. Multivariate Analysis

The PCA analysis evidenced the formation of three highly independent groups according to the distribution of the samples, where the shoot and root samples were separated by PC1, which explained 71.6% of the variation in the samples. Moreover, PC2 only separated the root samples according to the Cu treatment, explaining 9.1% of the total experimental variability. The first group included the samples of shoots both with and without Cu treatment, the second group collected the control root samples, while the third group included the root samples supplied with Cu (Figure 3).

### 2.4. Differential Expression Analysis

The differential expression analysis, which compared the control condition with the Cu addition, showed a high contrast in the response of the root organs regarding the shoots. In roots, a total of 7386 genes were differentially expressed, of which 3558 were up-regulated and 3828 down-regulated. Noticeably, in shoots, only 36 genes were differentially expressed, of which 13 were up-regulated and 23 down-regulated. A gene ontology analysis was performed with all differentially expressed genes (DEG) in the plant. This analysis classified the genes based in their functions on three general groups: (i) molecular function, (ii) biological process, and (iii) cellular component. Within the first group, only “structural molecule activity” presented a higher number of up-regulated than down-regulated genes. The other categories showed a higher number of down-regulated genes, highlighting “catalytic activity” and “transporter activity.” The genes associated with biological processes presented a higher number of down-regulated genes in all the subgroups. These results were evidenced mainly in the categories of “cellular process,” “metabolic process,” and “localization.” Finally, the group of cellular components concentrated the largest number of DEGs, where all the categories had more down-regulated than up-regulated genes (Figure 4).

A second analysis was performed to group the DEGs according to the class of protein to which they belong. According to the protein analysis through evolutionary relationships (PHANTER) classification system, the largest number of genes up-regulated in *I. cylindrica* corresponded to “translational protein,” “metabolite interconversion enzyme,” “protein modified enzyme,” “cytoskeletal protein,” and “transporter.” Otherwise, the down-regulated genes were a larger group than the up-regulated, with “metabolite interconversion enzyme” and “translational protein” being the most abundant (Figure 5). The search of DEGs in *I. cylindrica* shoots in the SwissProt database yielded only one up-regulated gene; cysteine-rich receptor-like protein kinase 10. Among the down-regulated gene group were noticeable the protein sulfur deficiency-induced 1, phosphate transporter PHO1-3, inorganic phosphate transporter 1–11, vacuolar iron transporter homolog 1, pectin acetylesterase 7, and bidirectional sugar transporter Sweet3a (Table 3). In the case of DEGs explored in root organs, many genes with described functionality were obtained, both for up-regulated and down-regulated genes (Table 4). We focused on those genes involved in Cu and metal tolerance response processes. Among them are superoxide dismutase (SOD) [Cu-Zn], SOD [Cu-Zn] 2, and SOD [Fe], mainly associated with the response against oxidative stress. Probable copper-transporting ATPase HMA5 and Hephaestin-like protein were also identified, both involved in the process of Cu ion transport. Additionally, a great number of genes involved in the cytoskeleton conformation (over 70) were up-regulated, with the most abundant being actin-1, actobindin-A, coactosin, and tubulin beta chain. On the contrary, among the down-regulated genes, several genes were involved both in the Cu and metal tolerance response and in the cytoskeleton conformation, namely: SOD [Mn] mitochondrial; extracellular SOD [Cu-Zn]; Cu transport protein ATOX1; Actin, muscle; Actin; Actin-1; and Actin-87E (Table 4).

## 3. Discussion

High levels of Cu affect key processes in plant development and growth, including the uptake of micro- and macronutrients, the reduction of pigment contents altering the photosynthesis process, and root expansion [30,31]. However, some plant species have a wide range of mechanisms to deal with metal stress and stop or at least diminish its deleterious effects on cellular processes [32]. Our study evidenced a noticeable difference in the transcriptomic response of root organs in comparison to shoot tissues in *I. cylindrica* growing under Cu stress conditions. Our results support that the root and rhizome are the main organ responsible for the tolerance to toxic Cu levels. Moreover, the PC analysis evidenced the homogeneity in the root organs according to the Cu treatment, which allowed it to separate into two clearly defined groups for Cu supply and control condition. This also supports the further homogeneity in the shoot samples irrespective of the Cu treatment (Figure 2 and Figure 3). These results are in agreement with previous report by Vidal et al. [20], where high concentrations of Cu (>300 mg kg^−1^ DW) were reported in root organs of *I. cylindrica*, compared to values about 10-fold lower in shoot tissues. The Cu uptake and accumulation firstly occurs at a great magnitude in the root organs, therefore producing a low rate of translocation to shoots [33,34]. Previous reports on *Elsholtzia splendens* and Bambusoideae plants showed that the majority of absorbed Cu is accumulated nearby the root rhizodermis as a layer [35,36]. Meanwhile, other studies in *Oryza sativa* and *Commelina communis* evidenced a major Cu accumulation in the vascular tissues at the root level [37,38].

Here, DEGs in shoots suggested that high levels of Cu in *I. cylindrica* rhizosphere generate principally a down-regulation of genes involved in transport processes. Among them is the gene encoding for protein sulfur deficiency-induced 1, involved in the utilization of stored sulfate under sulfur-deficient conditions [39]. Additionally, genes encoding for phosphate transporter PHO1–3, inorganic phosphate transporter 1–11, were down-regulated. In the case of genes encoding for phosphate transporters, plants can induce the overproduction of these membrane proteins to facilitate the introduction of inorganic phosphate (Pi) from the rhizosphere and thus satisfy biological demand during signaling and energy processes [40]. In addition, gene encoding for vacuolar iron transporter homolog 1 was also down-regulated in the Cu treatment. These results suggest that excess of Cu in plants produces an alteration in the transport of other essential nutrients mainly in the shoots. Noticeably, in the Cu time exposure here used (21 days) typical deleterious damages (such as chlorosis, necrosis, and death, among others) associated to nutrient deficiency were not observed. The architecture of this species could be in part responsible for this visual observation, since the presence of rhizomes can allow the preferential Cu accumulation in underground tissues and concomitantly act as a source of nutrients for the functioning of the shoots. However, this hypothesis must be further studied, including a series of chemical, morphological, omics, and ultrastructural tools to decipher whether the acquisition of nutrients by the plant increased or decreased under Cu stress.

Exposure to toxic Cu levels induces the formation of reactive oxygen species (ROS) inside plant cells, which leads to an imbalance in red-ox homeostasis [41,42]. The accumulation of ROS generates increasing oxidation of tissue components that can switch on a multitude of damaging effects, including injury of membranes, nucleic acids, proteins, and carbohydrates [43], finally causing the death of the organism [6,44]. However, plants can face these negative effects through intracellular mechanisms, being the antioxidant mechanisms the first line of action against this stress.

A response of down-regulated and up-regulated genes with SOD function was evidenced in the root transcriptome of *I. cylindrica* when the Cu treatment was used in the substrate. The presence of SOD is a common response to oxidative stress in all the plants, where these antioxidant enzymes act in the control over cellular level of free radicals [45,46], decreasing the cellular damage caused in this case by the Cu ions excess and alleviating Cu toxicity [47]. The down-regulated genes here found were the SOD [Mn], which codifies for an enzyme with mitochondrial localization, and the extracellular SOD [Cu-Zn]. Contrastingly, the SOD up-regulated genes identified by Cu addition were the cytosolic SOD [Cu/Zn] and the chloroplastic SOD [Fe] [48]. These results suggest the presence of an efficient mechanism of regulation involving both an intra- and extra-cellular level, mainly associated with the Cu localization, which could be mediated by the damage generated in each individual organelle. Meanwhile, contrarily to the regulation of ion transporter genes that occurred in shoots, in *I. cylindrica* root was evidenced a variety of these up-regulated genes, and only one down-regulated gene was identified. Genes encoding for Hephaestin-like protein and ABC transporter C family member 8 were part of this group. Hephaestin-like protein may function as a ferroxidase and may be involved in Cu transport and homeostasis; moreover, ABC transporter belongs to a large protein family that in plants are involved in the transport of a range of essential and PTEs (as Cu^+2^, Zn^+2^, Cd^+2^, Pb^+2^) across cell membranes [7,34,49]. The Probable copper-transporting ATPase HMA5, in turn, plays a key role in transmembrane Cu transport and is specifically induced by Cu in plants [50]. Moreover, studies in *Oryza sativa* demonstrated that this gene is involved in the loading of Cu to the xylem in roots and other organs [51], playing a key role in the Cu accumulation in root organs, which agrees with other observations by Vidal et al. [20]. The joint up-regulation of these genes demonstrated their importance in the Cu detoxification process in the plant roots, since these transporters take Cu ions from the soil, further distributing among plant organs and finally compartmentalize into subcellular organelles in the cell [52].

Among the DEGs in the root, an abundant group was formed by genes associated with the conformation and regulation of the cytoskeleton. This battery of genes included: actin, actin-1, actin-87E, actin related protein 2, and actobindin-A, among more than 100 other genes involved in structural function in the cell. Actin is the protein constituting the cytoskeleton microfilaments. Exposure to higher Cu levels can interfere with tyrosine phosphorylation of actin molecules, which is one of mechanisms of actin filaments depolymerization [53]. Meanwhile, the cytoskeletal microtubules composed are dynamic and their polymerization/depolymerization cycles are affected by many factors, such as temperature, drugs, metallic ions presence, or even other compounds able to increase the ROS as herbicides [54,55,56]. Studies about the effect of metals on the conformation of the mussel hematocyte cytoskeleton suggested that metals such as Cd and Cu could directly bind to cytoskeleton proteins followed by denaturation [57]. Another study by Jia et al. [58] showed how the Cu-oxide nanoparticles altered the cytoskeleton conformation in roots of *Arabidopsis thaliana*. In this case, the Cu-nanoparticles could conjugate with actin protein, inhibiting actin polymerization and concomitantly promoting its depolymerization. These antecedents could explain the high number of DEGs associated with the cytoskeleton, which would initially be acting as a target site of Cu-binding and then would be fragmented. The latter can generate the need for resynthesize these molecules and to ensure the structural and functional integrity of the cell. In this way, Xu et al. [59] concluded that although the cytoskeleton is primarily responsible for Pb/Cu-associated toxicity in plants, it can also be responsible for the tolerance. To date, there are few studies examining the behavior of the cytoskeleton against high concentrations of Cu in plants, but this behavior can represent a target mechanism to search in potential plants oriented to the remediation by using PTE-phytostabilization. For this reason, it is necessary to continue investigating and integrating molecular and microscopic techniques to elucidate whether the cytoskeleton is indeed acting as an (indirect) mechanism of Cu tolerance, but also representing an efficient mechanism for the PTE stabilization in root organs of metallophyte.

## 4. Materials and Methods

### 4.1. Plant Material and Experimental Design

Rhizomes of *I. cylindrica* were collected in the Puchuncaví Valley (Central Chile, 32°4603000 S, 71°2801700 W), Valparaíso Region, Chile, about 1.5 km from the Ventanas Cu-smelter. The rhizomes were cut and disinfected with 2% *w/v* chloramine-T solution for 5 min, then rinsed thoroughly with distilled water. Sterile-inert substrate was used for sprouting rhizomes, which was composed of sand and vermiculite (9:1; *v/v*). Plantlets were grown in a greenhouse for 6 months at 16/8 h light/dark photoperiod, at 25 ± 3/15 ± 3 °C day/night temperatures, and watering with sterile distilled water. The substrate was washed and sterilized by autoclaving per three consecutive days and air-dried for 24 h. After sterilization, the sand/vermiculite mixture was supplemented with the equivalent to 200 mL of a solution of 5.9 g L^−1^ of CuSO_4_·5H_2_O, which was allowed to equilibrate for 2 weeks at room temperature, as described by Aponte et al. [2]. This substrate represented the treatment with a nominal equivalent of 300 mg Cu kg^−1^. For the control treatment, 200 mL of distilled water were added to the substrate and let to equilibrate for 2 weeks. The assay compiled treatments with 300 mg Cu kg^−1^ substrate and controls without Cu addition, which were performed in pots with three plants per pot and four experimental units per treatment (*n* = 4). A basal fertilization of 18, 8, and 8 mg kg^−1^ of N, P, and K, respectively, was applied to all plots using a commercial fertilizer (Vitasac 18-8-8, Anasac Ambiental S.A., Santiago, Chile). After 6 months of growth, the plants were incorporated to the described treatments. To measure the genetic response associated to Cu tolerance mechanisms in *I. cylindrica*, the controls and treatments were grown in a greenhouse under the conditions described above for 21 days after the Cu application, based on previous reports by Vidal et al. [20]. After this period, the plants were harvested, separated into shoots and root organs (roots and rhizomes), weighed, and immediately frozen at −80 °C for further RNA extraction.

### 4.2. Cu Localization in Plant Tissues

In order to localize the Cu-bound to the shoot and root tissues, plants with 21 days of growth were observed by Variable Pressure Scanning Electron Microscope (VP-SEM), with transmission module STEM SU-3500 (Hitachi, Tokyo, Japan). The presence of Cu was verified by Energy Dispersive X-Ray Spectrometer Detector (EDX), QUANTAX 100 (Bruker, Karlsruhe, Germany) with BSE detector in transversal sections of shoot and root (rhizome). The analyses were carried out in the Scientific and Technological Bioresource Nucleus (BIOREN, Universidad de La Frontera, Temuco, Chile). Additionally, root and shoot tissues were dried, grounded, converted into ashes, and digested in a mix of dH_2_O:concentrated HCl:concentrated HNO_3_, 8:1:1 *v*:*v*:*v*, and the Cu was measured by atomic absorption spectroscopy (AAS; Unicam SOLAAR, mod. 969, Cambridge, UK).

### 4.3. RNA-Extraction, Sequencing, and De novo Transcriptome Assembly

Total RNA was extracted from frozen samples of shoots and roots using E.Z.N.A^®^ Total RNA Kit (Omega Bio-Tek, Norcross, GA, USA), according to the manufacturer’s instructions. A total of 16 RNA extractions (two tissues [shoot and root] × two treatments [with and without Cu] × four biological replicates) were performed, and the products were sent to Macrogen Inc, Korea, for sequencing. The quality and quantity in all the RNA extractions were checked using an Agilent Technologies 2100 Bioanalyzer (USA), where an RNA Integrity Number (RIN) value greater than or equal to 7 was adequate to perform the cDNA library construction. For each total RNA sample, a library of cDNA was constructed using TruSeq stranded kit (Illumina Inc., San Diego, CA, USA). Fifteen libraries were sequenced in an Illumina NovaSeq 6000 platform for 150 cycles in paired-end mode. The resulting FASTQ files containing the sequencing output were processed with NGSQC Toolkit [60] to evaluate the overall quality of the reads. This tool was also used to remove adaptors and reads with a low content of high-quality bases, where sequences with less than 70% of high-quality bases were removed from downstream analyses, being defined as high-quality the bases with a q-score higher than 30. Due to the lack of a reference genome for *I. cylindrica*, we used a de novo approach using Trinity v2.9.1 for reconstructing the transcriptome. It uses de Bruijn graphs to reconstruct transcripts and their variants using an extensive K-mer search strategy [61]. To assess assembly integrity, a set of conserved orthologous contained in the Embryophyta OrthoDBv9 database were queried by similarity search to *I. cylindrica* assembled transcripts using BUSCO [62].

### 4.4. Differential Gene Expression Analysis

The relative abundance of each sample was calculated using RSEM v1.2.26 [63] and merged in a matrix. It was analyzed using Bioconductor package DESeq2 [64], where a significant variation in gene expression was defined by a False Discovery Rate (FDR) lower than 0.01 and a minimum fold change (FC) of 4.

### 4.5. Transcriptome Annotation Analysis

All resulting genes were aligned into the UniProt/SwissProtKB database using BLAST+ with an e-value of 1 × 10^−10^ as the threshold. Functional annotation and ontology assignments were performed using the PANTHER classification system [65] with gene lists obtained from blast results (top hit) aligned with reference proteome database (version 2018_4) from EMBL.

## 5. Conclusions

Differential expression analysis using the de novo transcriptome of *Imperata cylindrica* plants growing under Cu stress generated the identification of a large number of genes involved in the Cu tolerance response, especially in root organs. Our results strongly support the importance of the root organs of *I. cylindrica* as the main responsible for Cu tolerance and as a sink that ultimately produces the metal bioaccumulation. Moreover, some genes with structural function in the cytoskeleton are also proposed as an interesting mechanism for reducing the Cu toxicity, presumably through the Cu-binding to actin microfilaments and microtubules. The latter is especially important if phytostabilization programs are planned using Cu-tolerant metallophytes, in which case *I. cylindrica* emerges as an ideal candidate. Nevertheless, more in-deep studies are necessary to validate this hypothesis.

## Figures and Tables

**Figure 1 plants-10-00357-f001:**
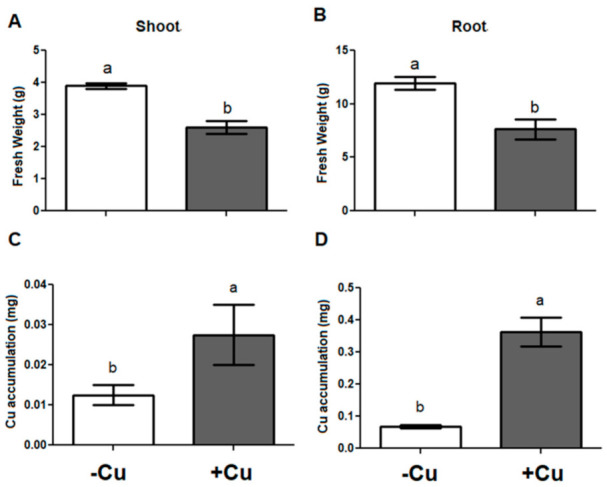
Biomass production (g fresh weight per experimental unit) of *Imperata cylindrica*: (**A**) shoots biomass, (**B**) root biomass; and Cu accumulation in (**C**) shoot and (**D**) root. –Cu, treatment without Cu applied, and +Cu treatment with addition of 300 mg Cu kg^−1^ substrate. Different lowercase letters indicate differences according to t-tests for independent samples (*p* < 0.05). Data are expressed as mean ± SE, *n* = 4.

**Figure 2 plants-10-00357-f002:**
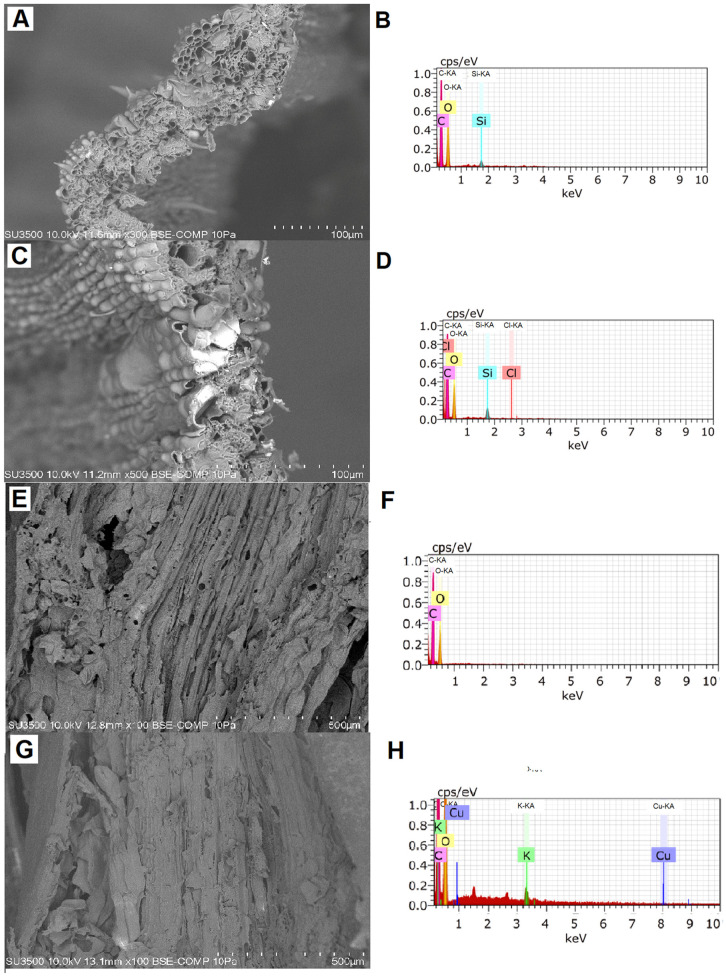
Scanning electron microscopy (SEM) in transversal sections of *Imperata cylindrica* shoot (**A**) without Cu addition, (**B**) Energy-dispersive X-ray (EDX) spectra without Cu addition, (**C**) SEM of shoot with Cu addition (300 mg kg^−1^), and (**D**) EDX spectra with Cu addition. SEM in longitudinal sections of *I. cylindrica* root (**E**) without Cu addition, (**F**) EDX spectra without Cu addition, (**G**) SEM of root with Cu addition (300 mg kg^−1^), and (**H**) EDX spectra with Cu addition.

**Figure 3 plants-10-00357-f003:**
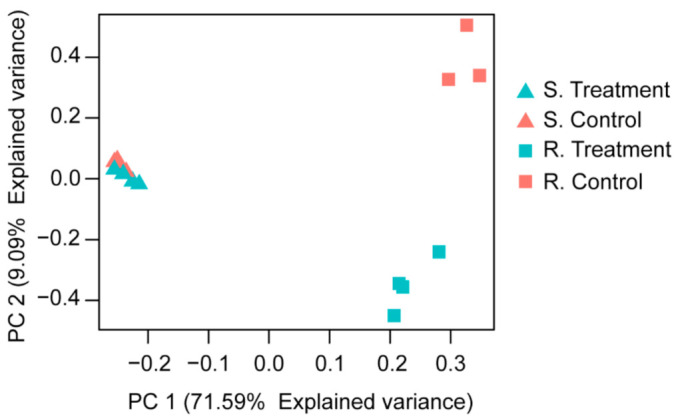
Principal Component Analysis of *Imperata cylindrica* in shoots and roots samples. The control treatment was established without Cu addition, while the treatment consisted in the addition of 300 mg Cu kg^−1^ substrate.

**Figure 4 plants-10-00357-f004:**
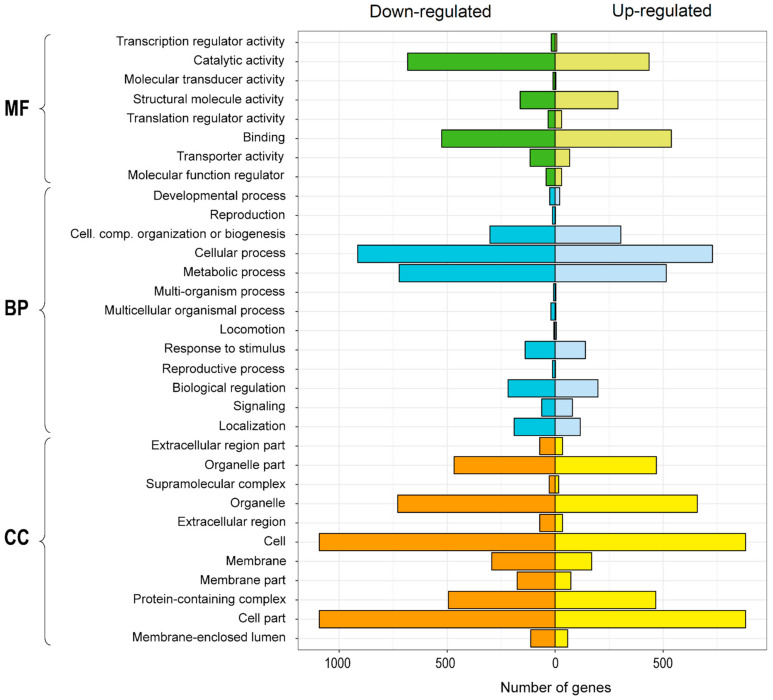
Gene ontology distribution in de novo transcriptome of *Imperata cylindrica* according to level 1 categories: Cellular Component (CC), Biological Process (BP), and Molecular Function (MF). Genes are differentially expressed in treatment compared to control conditions.

**Figure 5 plants-10-00357-f005:**
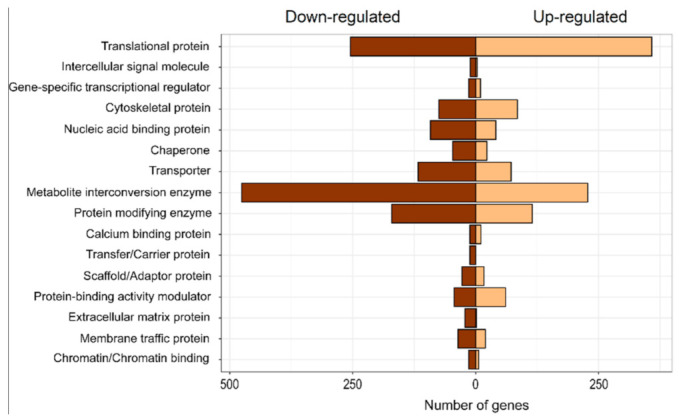
Distribution of differentially expressed genes according to the protein analysis through evolutionary relationships (PHANTER) classification system in plants of *Imperata cylindrica*. The bars show the number of genes encoding for each functional class. Genes are differentially expressed in treatment with Cu addition (300 mg Cu kg^−1^ substrate) compared to control condition without Cu addition.

**Table 1 plants-10-00357-t001:** De novo transcriptome assembly of *Imperata cylindrica* by RNA-Seq.

Metric *I. cylindrica* Transcriptome	
Trinity transcripts	200,521
Trinity “genes”	113,137
GC (%)	50
Contig N50 (bp)	1918
Median contig length (bp)	978
Average contig (bp)	1377
Total assembled bases	276,047,337
Blastx Hit SwissProt	(58%) 117,238

**Table 2 plants-10-00357-t002:** Validation of the de novo transcriptome assembly of *Imperata cylindrica*.

Summary of Benchmarking Universal Single-Copy Orthologs (BUSCO) Statistics		
Complete BUSCOs	1259	91.5%
Complete and single-copy BUSCOs	560	40.7%
Complete and duplicated BUSCOs	699	50.8%
Fragmented BUSCOs	85	6.2%
Missing BUSCOs	31	2.3%
Total BUSCO genes *	1375	100.0%

* Database: embriophyte_odb10.

**Table 3 plants-10-00357-t003:** Differential expressed genes (DEGs) in shoots of *Imperata cylindrica* plants growing under Cu stress conditions (300 mg kg^−1^) with respect to the control condition (without Cu).

Gene Encoding for	Ontology	FPKM Control	FPKM Treatment	Fold Change (FC)	*p*-Value
Cysteine-rich receptor-like protein kinase 10	MF: ATP binding (GO:0005524)	0	233.66	10.61	7.80 × 10^−22^
Protein SULFUR DEFICIENCY-INDUCED 1	BP: regulation of sulfur utilization (GO:0006792)	76.03	1.06	−5.11	9.00 × 10^−5^
Protein SULFUR DEFICIENCY-INDUCED 1	BP: cellular response to sulfur starvation (GO:0010438)	1817.86	229.38	−7.69	2.70 × 10^−11^
Phosphate transporter PHO1-3	PB: phosphate ion transport (GO:0006817)	98.79	11.16	−4.92	1.80 × 10^−4^
Inorganic phosphate transporter 1–11	BP: phosphate ion transport (GO:0006817)	2162.05	269.6	−4.69	4.30 × 10^−4^
Vacuolar iron transporter homolog 1	BP: intracellular sequestering of iron ion (GO:0006880)	44.07	0	−7.57	5.70 × 10^−11^
GDP-L-galactose phosphorylase 2	MF: hydrolase activity (GO:0016787)	9153.18	1541.18	−6.73	1.80 × 10^−8^
Pectin acetylesterase 7	BP: cell wall organization (GO:0071555)	211.39	13.83	−4.44	1.10 × 10^−3^
Bidirectional sugar transporter SWEET3a	MF: sugar transmembrane transporter activity (GO:0051119)	104.55	3.94	−4.58	6.40 × 10^−4^
RNA exonuclease 4	PB: rRNA processing (GO:0006364)	53.93	5.02	−4.65	4.9 × 10^−4^

FPKM = Fragments per kilobase of transcript per million mapped reads. FC = Fold Change, based on the log_2_ FC value.

**Table 4 plants-10-00357-t004:** Differentially expressed genes (DEG) related to Cu-tolerant mechanisms in roots of *Imperata cylindrica* plants growing under Cu stress conditions (300 mg kg^−1^) with respect to the control condition (without Cu).

Gene Encoding for	Ontology	FPKM Control	FPKM Treatment	Fold Change (FC)	*p*-Value
Superoxide dismutase [Mn], mitochondrial	PB: removal of superoxide radicals (GO:0019430)	94.74	0	−4.08	6.79 × 10^−4^
Extracellular superoxide dismutase [Cu-Zn]	PB: removal of superoxide radicals (GO:0019430)	77.17	0	−3.82	1.62 × 10^−3^
Superoxide dismutase [Cu-Zn]	PB: removal of superoxide radicals (GO:0019430)	0.55	25.13	3.90	1.3 x10^−3^
Superoxide dismutase [Cu-Zn] 2	PB: removal of superoxide radicals (GO:0019430)	6.68	82.63	4.36	2.5 × 10^−4^
Superoxide dismutase [Fe]	MF: superoxide dismutase activity (GO:0004784)	12.32	82.56	4.17	5.0 × 10^−4^
Copper transport protein ATOX1	BP: cellular copper ion homeostasis (GO:0006878)	47.45	0	−3.53	4.07 × 10^−3^
Hephaestin-like protein	BP:copper ion transport (GO:0006825)	2.68	36.63	5.17	9.6 × 10^−6^
Probable copper-transporting ATPase HMA5	BP:detoxification of copper ion (GO:0010273)	1.72	23.09	3.95	1.1 × 10^−3^
Protein NBR1 homolog	BP:protein transport (GO:0015031)	9.18	100.88	7.02	5.0 × 10^−10^
ABC transporter C family member 8	BP: transmembrane transport (GO:0055085)	5.41	34.58	3.88	1.3 × 10^−3^
Actin, muscle	CC: cytoskeleton (GO:0005856)	92.68	3.17	−4.37	2.34 × 10^−4^
Actin	CC: cytoskeleton (GO:0005856)	7459.58	1584.51	−4.14	5.38 × 10^−4^
Actin-1	CC: cytoskeleton (GO:0005856)	162.63	18.93	−3.80	1.72 × 10^−3^
Actin-87E	CC: cytoskeleton (GO:0005856)	1441.51	0.49	−6.70	3.39 × 10^−9^
Actin-1	CC: cytoskeleton (GO:0005856)	13.44	218.38	4.98	2.2 × 10^−5^
Actin-related protein 2	MF:actin binding (GO:0003779)	0.55	46.29	6.28	4.0 × 10^−8^
Actobindin-A	MF:actin binding (GO:0003779)	5.49	234.97	6.19	6.6 × 10^−8^

FPKM = Fragments per kilobase of transcript per million mapped reads. FC = Fold Change, based on the log_2_ FC value.

## Data Availability

Data available on request due to restrictions eg privacy or ethical. The data presented in this study are available on request from the corresponding author. The data are not publicly available due to the size and format of the files.

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
