# Peer review of "Main Molecular Pathways Associated with Copper Tolerance Response in Imperata cylindrica by de novo Transcriptome Assembly"

_plants, 2021, doi:10.3390/plants10020357_

Round 1

Reviewer 1 Report

The study carried out by Vidal et al., is a nice contribution to understand the mechanismes involved in Cu tolerance in a metallophyte  plant, Imperata cylindrica, by using transcriptomic approaches. The authors  performed transcriptomic  analyses of shoots and roots, with and without Cu supply and differentially expressed genes (DEGs) in both organs were analyzed and grouped in different categories.

I suggest the authors to analyze some genes related with the most interesting categories, such as cytoskeleton and metal transporter to validate the results obtained in this manuscript.

It would be very helpful to analyze SOD activity in Imperata cylindrica with and without Cu treatment to corroborate the results obtained.

Actin polymerization can be disturbed by ROS and RNS-dependent PTMs, although this issue has not been discussed in the manuscript (see work by Rodríguez-Serrano et al., Journal of Experimental Botany 65,4783 -4793, 2014).

Author Response

I suggest the authors to analyze some genes related with the most interesting categories, such as cytoskeleton and metal transporter to validate the results obtained in this manuscript.

R: Thank you very much for your comments, they certainly enrich our work. The validation of the main genes highlighted in our work it was always part of our goals. However, at the present we are unable to check some genes (for instance through qt-PCR) because or labs and the entire University is closed by the complex situation in our city for the covid-19 pandemic. In fact, Temuco is now in the third quarantine and from past march ’20 with all our facilities closed. Nevertheless, in our study we have considered the analysis including other tools to describe the in situ capability of I. cylindrica to cope with the Cu toxicity, including the use of selected AM fungal strains, so we hope to include this issue in our next assay.

It would be very helpful to analyze SOD activity in Imperata cylindrica with and without Cu treatment to corroborate the results obtained.

R: Indeed, this enzymatic analysis will provide solidity to our work and will be carried out post-Covid19 pandemic included in our next assay. In fact, we have previously optimized a series of techniques to describe the oxidative response through enzymatic and non-enzymatic approaches, which can be complimentary to our previous studies regarding the description of antioxidant profiles of phenolic compounds of I. cylindrica plants growing under Cu stress (please, see Vidal et al., 2020, Plants 9:1397) and also the possible modifications by the AM symbiosis.

Actin polymerization can be disturbed by ROS and RNS-dependent PTMs, although this issue has not been discussed in the manuscript (see work by Rodríguez-Serrano et al., Journal of Experimental Botany 65,4783 -4793, 2014).                        

R: Certainly, the aforementioned article is very interesting and among its results highlights the damage in the conformation of the cytoskeleton of Arabidopsis cells exposed to the herbicide 2,4-Dichlorophenoxyacetic acid. Since the production of ROS is mostly unspecific, we taken this suggestion in the discussion to support this important point. Thank you for your suggestions.

Reviewer 2 Report

The manuscript presents studies of Imperata cylindrica, a monocot with potential in Cu phytoremediation. Authors performed expression analysis on vegetatively propagated plants exposed to high Cu compare to control low Cu conditions. The analysis showed interesting difference between roots and shoots pointing the role of roots in management of excess of Cu. The results in roots exposed to Cu are significant for our understanding of Cu homeostasis but also may help to find the mechanisms useful for phytoremediation of Cu polluted soils. Unfortunately, authors did not used the potential that make sequencing analysis so powerful. Authors need to make a more comprehensive summary figure of their results, e.g. graphic figure presenting the most significant DEGs, their function, cellular localization and role in Cu homeostasis (e.g. uptake, compartmentalization, root-to-shoot transport etc.). Authors omit in the discussion interpretation of significant Cu homeostasis genes e.g. is HMA5! that was just mentioned, but it’s role was not discussed - HMA5 was show to play a direct role in root-to-shoot Cu translocation (e.g. shown in OsHMA5). I believe other transporters that play a role in Cu uptake and root-to-shoot translocation should be extensively described and their expression in I. cylindica discussed in relation to plant metallophyte properties. Additionally, authors need to attach or refer to full list of DEGs highlighting those that are known to play a role in Cu or nutrient homeostasis.

I was wondering if among roots DEGs there are some homologs of phytochelatin genes or gluthation (or other ROS scavenging)?

The other significant problem is that expression analysis in this manuscript is not followed up by any experiments confirming the results or making even experimental effort to back up expression based speculation (in discussion/conclusion):

Authors need to:

  1. confirm few most significant DEGs for I. cylindrica properties as metallophyte (related to cytoskeleton, shoot Pi transporters etc….)
  2. Line 215-217 authors make conclusion but did not show direct evidence that there is an excess of Cu, or nutrients deficiency in shoots of plants grown in Cu treatment. To make such a statement and improve significantly connection between gene expression and plant physiology authors have to compare nutrient composition of treated and untreated plants (by e.g. ICP-MS – for Cu but maybe also Pi, Fe, Zn, Ca…). Moreover, author suggest “block in acquisition” when they describe shoots. That phenotype would be related to uptake of nutrients by roots, the shoots respond to decreased root-to-shoot translocation of elements. The translocation of Cu to the shoots might be in fact affected by high Cu in roots (authors should search literature to check that possibility in monocots). Nevertheless, downregulation of nutrient transporters (especially those expressed in deficiency) impose that plant do not suffer nutrient deficiency (contrary to what authors suggest)!
  3. Table 4 present result but it is not clear why those (FC threshold? or importance for Cu?, or maybe by annotation/known function?). Authors should specify what is in table 4 and why they choose to show only those. Anyway, both up and down regulated genes list is needed (in SI, some down- regulated genes related to Cu homeostasis should be also listed in main manuscript).
  4. The speculation about cytoskeleton role in Cu homeostasis is not well supported. I really like idea of cytoskeleton role in Cu detoxification, what is needed are experiments showing that something happens to cytoskeleton under Cu treatment (gene expression sometimes might not be relevant to the plant phenotype). One test that can be done is actin/tubulin immunolabeling and confocal microscopy of Cu treated roots compare to control. E.g. (FYI I am not an author or collaborator in this paper) -> Dyachok, Julia & Paez, Ana & Yoo, Cheol-Min & Karuppaiah, Palanichelvam & Blancaflor, Elison. (2015). Fluorescence Imaging of the Cytoskeleton in Plant Roots. 10.1007/978-1-4939-3124-8_7. Optimally (but optionally), this should be related to Cu localization in cell (subcellular level –  g. by using TEM with EDS analysis).

Additionally, English language seems to be an issue in some part of the manuscript. There are many syntax errors. I had a lot of problems with understanding context of the results, mostly because too long sentences. I would suggest to use rather simple structure. I tried to make some comments but as non-native speaker I would advise to ask for help of native speaker collaborators – it will make paper better.

Author Response

The manuscript presents studies of Imperata cylindrica, a monocot with potential in Cu phytoremediation. Authors performed expression analysis on vegetatively propagated plants exposed to high Cu compare to control low Cu conditions. The analysis showed interesting difference between roots and shoots pointing the role of roots in management of excess of Cu. The results in roots exposed to Cu are significant for our understanding of Cu homeostasis but also may help to find the mechanisms useful for phytoremediation of Cu polluted soils. Unfortunately, authors did not used the potential that make sequencing analysis so powerful. Authors need to make a more comprehensive summary figure of their results, e.g. graphic figure presenting the most significant DEGs, their function, cellular localization and role in Cu homeostasis (e.g. uptake, compartmentalization, root-to-shoot transport etc.). Authors omit in the discussion interpretation of significant Cu homeostasis genes e.g. is HMA5! that was just mentioned, but it’s role was not discussed - HMA5 was show to play a direct role in root-to-shoot Cu translocation (e.g. shown in OsHMA5). I believe other transporters that play a role in Cu uptake and root-to-shoot translocation should be extensively described and their expression in I. cylindica discussed in relation to plant metallophyte properties. Additionally, authors need to attach or refer to full list of DEGs highlighting those that are known to play a role in Cu or nutrient homeostasis.

R: First at all, thank you very much for your positive comments about our paper. We are expecting about the actual possibilities for the use of I. cylindrica for in field programs of bioremediation in large areas Cu polluted in north and central Chile. We have some background regarding some exclusion mechanisms in this species (Meier et al., 2012, Ecotox Environ Safe) and the capability to establish associations with efficient AM fungi. All those aspects are now being considered for a complete frame to explain the capabilities for using this species. Some of our new evidence was previously included in Vidal et al. (2020 Plants), maybe giving some other relevant pieces for the understanding of the complex mechanisms of this species to cope with the Cu. Therefore, we feel that make a graphical support is currently advantageous without the in-field knowledge and the use of other tools for the ultralocalization of Cu in the plant tissues. However, is an excellent idea when the above-mentioned results be available. Regarding the second comment, you have reason about the interpretation of other significant Cu genes. We are now submitting a most in deep analysis of the relevant genes (including f.i. HMA5), but curiously the interpretation remains similar because the enormous differences between the changes at root level compared to the others occurring at shoot level. Maybe, with other proteomics or metabolomics approaches could be possible a most accurate explanation. But we feel that starting from a De novo transcriptome our results are still relevant and novel. Finally, regarding the list of DEGs, we have modifier the results and tables, including the relevant DEGs as you suggest. Moreover, we are including the date set as supplementary material if possible, or alternatively making available for other authors if requested.

I was wondering if among roots DEGs there are some homologs of phytochelatin genes or gluthation (or other ROS scavenging)?

R: Thank you for this comment. Certainly, based on antecedents we can hypothesized that some homologs of phytochelatins or metallotionein must be up-regulated, However, according with our results and the analysis of DEGs were not identified as one of the main mechanisms of response. Perhaps, considering other criteria for the exclusion (a minor degree of p significante) the identification of some of these genes can be possible, but we have decided to use a high degree of confidence to select those most down or up regulated DEG. We thank your comment because in our next assay, and accounting with a most validated transcriptome we can focus on specific mechanisms, including other well-known as involved in the Cu-homeostasis.

The other significant problem is that expression analysis in this manuscript is not followed up by any experiments confirming the results or making even experimental effort to back up expression based speculation (in discussion/conclusion):

R: It is worth mentioning that this work is the continuation of a first study where Imperata cylindrica is studied under similar stress conditions and copper levels, mainly focused to establish the proper sampling time based in a wide range of phenotypical traits. The main findings of this study (Vidal et al, 2020) has been discussed in several points in our discussion, mainly reinforcing the clear preferential response in root organs to high Cu levels. Moreover, as previously mentioned to Rev1, we have as main some other assays to corroborate at genotypical level and in situ the capabilities of this metallophyte to be used in Cu-remediation. Thank you for your comment.

Authors need to:

1.-Confirm few most significant DEGs for I. cylindrica properties as metallophyte (related to cytoskeleton, shoot Pi transporters etc….)

R: Reinforcing, one of our objectives contemplated the validation of differentially expressed genes, especially those related to the response of the cytoskeleton to copper. However, this objective could not be achieved currently because the covid-19 pandemic paralyzed all our work for the last year. However, is one of our next steps to elucidate the properties of I. cylindrica at gene-expression level, and its use as part of technologies of bioremediation (using for instance AM fungi and other PGP microorganisms) in soils with environmentally high levels of Cu.

2.-Line 215-217 authors make conclusion but did not show direct evidence that there is an excess of Cu, or nutrients deficiency in shoots of plants grown in Cu treatment. To make such a statement and improve significantly connection between gene expression and plant physiology authors have to compare nutrient composition of treated and untreated plants (by e.g. ICP-MS – for Cu but maybe also Pi, Fe, Zn, Ca…). Moreover, author suggest “block in acquisition” when they describe shoots. That phenotype would be related to uptake of nutrients by roots, the shoots respond to decreased root-to-shoot translocation of elements. The translocation of Cu to the shoots might be in fact affected by high Cu in roots (authors should search literature to check that possibility in monocots). Nevertheless, downregulation of nutrient transporters (especially those expressed in deficiency) impose that plant do not suffer nutrient deficiency (contrary to what authors suggest)!

R: Thank you for your very constructive comment. We must clarify two aspects regarding your observation. Firstly, most of the phenotypical characterization was assumed in our previous study in Vidal et al. (2020), including the compartmentalization of Cu in roots and shoots, where was possible to observe the gross Cu accumulation is occurring in the root tissues, mainly rhizomes. This was also evidenced at microscopical level using EDX tools. Secondly, based in the above we have included some possible explanation about the role of rhizomes in this case. Being the typical structure for the propagation and the more recalcitrant structure of I. cylindrica in field conditions, we think this organ plays a main role not only accumulating the Cu but also serving as source of nutrients. This can allow the plant growth and other advantageous characteristics when is compared to other plants, even some monocotyledons plant species. As previously mentioned, we have a research on course where one of the next steps is the wide characterization of the genotypical response under in field conditions where a complete description of the physiological, metabolic and biochemical behavior of the plants is considered.

3.-Table 4 present result but it is not clear why those (FC threshold? or importance for Cu?, or maybe by annotation/known function?). Authors should specify what is in table 4 and why they choose to show only those. Anyway, both up and down regulated genes list is needed (in SI, some down- regulated genes related to Cu homeostasis should be also listed in main manuscript).

R: The fold change of the analysis of differentially expressed genes was 4, which was clarified in the footnote in Tables 3 and 4, please see the new ms. version. The genes that were selected for Table 4 were those associated with the response to copper stress, mainly those mentioned in the introduction (Line 66). Finally, based on your question, we have included some other interesting DEG not only up-regulated but also some other down-regulated. Please see the new version of Table 4.

4.-The speculation about cytoskeleton role in Cu homeostasis is not well supported. I really like idea of cytoskeleton role in Cu detoxification, what is needed are experiments showing that something happens to cytoskeleton under Cu treatment (gene expression sometimes might not be relevant to the plant phenotype). One test that can be done is actin/tubulin immunolabeling and confocal microscopy of Cu treated roots compare to control. E.g. (FYI I am not an author or collaborator in this paper) -> Dyachok, Julia & Paez, Ana & Yoo, Cheol-Min & Karuppaiah, Palanichelvam & Blancaflor, Elison. (2015). Fluorescence Imaging of the Cytoskeleton in Plant Roots. 10.1007/978-1-4939-3124-8_7. Optimally (but optionally), this should be related to Cu localization in cell (subcellular level –  g. by using TEM with EDS analysis).

R: Thank you very much for your support. In fact, as previously mentioned, we are in a more in deep research related to this metallophyte, where some tools of ultrastructural localization it is also planned. At our university we have some facilities to analyze at structural level the localization at semi-quantitative level which was reported in Vidal et al. (2020) using SEM(stem) coupled to EDX. Sure, we will use a more in deep ultralocalization tool for our next assay considering a first choice TEM-EDS that are available in other universities into our net.

Additionally, English language seems to be an issue in some part of the manuscript. There are many syntax errors. I had a lot of problems with understanding context of the results, mostly because too long sentences. I would suggest to use rather simple structure. I tried to make some comments but as non-native speaker I would advise to ask for help of native speaker collaborators – it will make paper better.

R: Thank you for your advice. As you can note we are not native English speakers, but we are made another deep revision on our manuscript to improve the writing in the whole document. Detailing, some of the changes were also suggested by the other colleagues. Please, see the new manuscript.

Reviewer 3 Report

The manuscript "Main molecular pathways associated with copper tolerance response in Imperata cylindrica by De novo transcriptome assembly" very interesting and up-to-date. There are a few minor errors in the text that should be corrected:

throughout the text, replace commas with dots for numerical results
- line 28 - change commas from red to black
- lines 80, 289 - convert all words to italics
- lines 83,84 - incorrect notation of commas and periods
Figure 1 - plant names in Latin are written in italics
- line 117 - unnecessary bold font
- line 235 - correct the order of citations in parentheses

Author Response

The manuscript "Main molecular pathways associated with copper tolerance response in Imperata cylindrica by De novo transcriptome assembly" very interesting and up-to-date. There are a few minor errors in the text that should be corrected:

R: Thank you very much for your positive comments.

Throughout the text, replace commas with dots for numerical results

- line 28 - change commas from red to black

R: The commas of the numbers are in black. Maybe was changed for editors previously to be sent to reviewers.

- lines 80, 289 - convert all words to italics

R: All subtitles, scientific names and Latin names were italicized (example: De Novo) through all the body text.

- lines 83,84 - incorrect notation of commas and periods

R: Commas and periods were corrected according to journal format. The numbers after the point were also eliminated since they were not significant numbers, with respect to the magnitude of the results.

Figure 1 - plant names in Latin are written in italics

R: The scientific name of the plant was changed to italics.

- line 117 - unnecessary bold font

R: The font is correct without bold letters. Was also reviewed in the entire text.

- line 235 - correct the order of citations in parentheses

R: The order of the citations was corrected.

Reviewer 4 Report

Authors have analyzed effects of copper on the transcriptome of Imperata cylindrica. Significant differences between roots and shoots have been observed. In order to interpret observed results, responsive gene products have been aligned to respective physiological functions.
As authors are lacking corresponding physiological and biochemical data they are discussing results by reviewing respective information extracted from earlier publications. Some of the cited data have been measured using other plant species, though. Nevertheless, conclusions presented by the authors sound convincing. Moreover, findings of the authors nicely underline importance of the cell skeleton in structuring the cytoplasm, and this way regulating metabolic activities.

I suggest accepting the manuscript in the present form. But to long sentences are resulting grammar errors. This applies to the manuscript as a whole. (see lines 43-47, for instance)

Author Response

Authors have analyzed effects of copper on the transcriptome of Imperata cylindrica. Significant differences between roots and shoots have been observed. In order to interpret observed results, responsive gene products have been aligned to respective physiological functions. As authors are lacking corresponding physiological and biochemical data they are discussing results by reviewing respective information extracted from earlier publications. Some of the cited data have been measured using other plant species, though. Nevertheless, conclusions presented by the authors sound convincing. Moreover, findings of the authors nicely underline importance of the cell skeleton in structuring the cytoplasm, and this way regulating metabolic activities.

R: Thank you very much for your positive comments about our study, conclusions, and new information. We are devoted to complement other aspects involved in the Cu tolerance by I. cylindrica as model of metallophyte. Therefore, in the near future effectively we will study the ultrastructural aspect of Cu immobilization and also using other omics tools to deeply understand the physiological and biochemical aspects regarding Cu homeostasis in this plant.

I suggest accepting the manuscript in the present form. But to long sentences are resulting grammar errors. This applies to the manuscript as a whole. (see lines 43-47, for instance)

R: Thank you very much for your review and good comments. We reviewed and improved grammar typos throughout the manuscript to provide a better understanding. Please, note that your comment was also given by reviewer 2.

Round 2

Reviewer 2 Report

Thank you for making some changes and respond, I perfectly understand COVID situation. I think that the paper is significant but with additional experiments I mentioned it would have much much higher impact...

Regarding nutrients deficiency. In my opinion authors cannot claim that: "These results suggest that excess of Cu in the plant can cause a decrease in the acquisition of other macro- and micro essential nutrients. Downregulation of nutrient transporters (nutrient deficiency indicators) impose that plant do not suffer nutrient deficiency! - I would expect those genes are upregulated in nutrient deficiency - so that is not enough evidence of actual shift in nutrient concentrations in plant. This, as authors are aware, is a very complex problem - the regulation of nutrient transporters depend on the nutrient concentration and other factors like the crosshomeostasis when one ion e.g. Fe can regulate other e.g. Zn uptake. So Cu might regulate those transporters or e.g. increased concentration of nutrient that is transported can do the same job - e.g. one can imagine that Cu may facilitate uptake of Pi and increased Pi downregulates its transporters... Simplification that Cu leads to downregulation of transporters so there is lower nutrient uptake of nutrients in plant is not justified. Simply, authors do not have evidence that there is "a decrease in nutrient acquisition". To justify it the direct ICP-MS analysis of nutrients concentrations is needed. I would encourage authors to correct that statement.

Nevertheless, I agree with authors that results are interesting and at this stage could be published as exploratory research. Although, I am looking forward to see followup.

Author Response

Responses to Reviewer 2, second round.

Thank you for making some changes and respond, I perfectly understand COVID situation. I think that the paper is significant but with additional experiments I mentioned it would have much much higher impact...

R: Thank you for your positive comments. I understand that some other additional experiments can give more impact to our study. As mentioned, previously we implemented other physiological and biochemical assays in a prospective study to define the Cu doses and the sampling time. However, key factors as the mentioned in your next comments will be positively covered in our forthcoming research.

Regarding nutrients deficiency. In my opinion authors cannot claim that: "These results suggest that excess of Cu in the plant can cause a decrease in the acquisition of other macro- and micro essential nutrients. Downregulation of nutrient transporters (nutrient deficiency indicators) impose that plant do not suffer nutrient deficiency! - I would expect those genes are upregulated in nutrient deficiency - so that is not enough evidence of actual shift in nutrient concentrations in plant. This, as authors are aware, is a very complex problem - the regulation of nutrient transporters depend on the nutrient concentration and other factors like the crosshomeostasis when one ion e.g. Fe can regulate other e.g. Zn uptake. So Cu might regulate those transporters or e.g. increased concentration of nutrient that is transported can do the same job - e.g. one can imagine that Cu may facilitate uptake of Pi and increased Pi downregulates its transporters... Simplification that Cu leads to downregulation of transporters so there is lower nutrient uptake of nutrients in plant is not justified. Simply, authors do not have evidence that there is "a decrease in nutrient acquisition". To justify it the direct ICP-MS analysis of nutrients concentrations is needed. I would encourage authors to correct that statement.

R: Your comments and assertions are completely correct. Thank you for your clarifications and suggestions about this key point. We take into consideration the need for a most deep characterization of the nutrient status of I. cylindrica plants under Cu stress. Moreover, considering that this is a De novo transcriptome, a most accurate description of a significant fraction of not annotated genes could be also an interesting source of information for a most complete description of other homeostatic processes regarding the Cu levels in the plant (consider that this species has not a reference genome yet). At this time, was not possible to store a significant quantity of plant tissue because the amount of fresh material to obtain a high-quality conserved RNA was also an unknown factor. So, we opted for ensure a maximal yield of RNA because our alternative for a correlated nutrient characterization is at the present using AAS, which usually requires several times more plant amounts than ICP techniques. After this explanation, we have performed some minor changes in the manuscript, avoiding the direct references to the nutritional status as effect of Cu supply:

The change was made in the lines 240-248 (in the track changes version):

These results suggest that excess of Cu in the plant produce an alteration in the transport of other essential nutrients mainly the shoots. Noticeably, in the Cu time exposure here used (21 days) typical deleterious damages (as chlorosis, necrosis, death, among others) associated to nutrient deficiency were not observed. Noticeably, the architecture of this species could be in part responsible for this visual observation since the presence of rhizomes can allow the preferential Cu accumulation in underground tissues and concomitantly be acting as a source of nutrients for the functioning of the shoots. However, this hypothesis must be further studied including a series of chemical, morphological, -omics and ultrastructural tools to decipher whether the acquisition of nutrients by the plant increased or decreased under Cu stress.

Nevertheless, I agree with authors that results are interesting and at this stage could be published as exploratory research. Although, I am looking forward to see followup.

R: Again, thank you for your positive feeling about our study. We also hope to complement our information regarding the use of I. cylindrica supporting its use in phytoremediation of Cu-polluted environments.